# Cheap and Quick: Efficient Vision-Language Instruction Tuning for Large Language Models

**Gen Luo**[13], **Yiyi Zhou**[12], **Tianhe Ren**[1], **Shengxin Chen**[1], **Xiaoshuai Sun**[12], **Rongrong Ji**[123*]

[1]Key Laboratory of Multimedia Trusted Perception and Efficient Computing,
Ministry of Education of China, School of Informatics, Xiamen University, 361005, P.R. China.
[2]Institute of Artificial Intelligence, Xiamen University, 361005, P.R. China.
[3] Peng Cheng Laboratory, Shenzhen, 518000, China.
{luogen,chenshengxin,rentianhe}@stu.xmu.edu.cn,
{zhouyiyi,xssun,rrji}@xmu.edu.cn

## Abstract

Recently, growing interest has been aroused in extending the multimodal capability of large language models (LLMs), *e.g.*, vision-language (VL) learning, which is regarded as the next milestone of artificial general intelligence. However, existing solutions are prohibitively expensive, which not only need to optimize excessive parameters, but also require another large-scale pre-training before VL instruction tuning. In this paper, we propose a novel and affordable solution for the effective VL adaption of LLMs, called *Mixture-of-Modality Adaptation* (MMA). Instead of using large neural networks to connect the image encoder and LLM, MMA adopts lightweight modules, *i.e.*, adapters, to bridge the gap between LLMs and VL tasks, which also enables the joint optimization of the image and language models. Meanwhile, MMA is also equipped with a routing algorithm to help LLMs achieve an automatic shift between single- and multi-modal instructions without compromising their ability of natural language understanding. To validate MMA, we apply it to a recent LLM called LLaMA and term this formed *large vision-language instructed* model as LaVIN. To validate MMA and LaVIN, we conduct extensive experiments under two setups, namely *multimodal science question answering* and *multimodal dialogue*. The experimental results not only demonstrate the competitive performance and the superior training efficiency of LaVIN than existing multimodal LLMs, but also confirm its great potential as a general-purpose chatbot. More importantly, the actual expenditure of LaVIN is extremely cheap, *e.g.*, only **1.4 training hours** with 3.8M trainable parameters, greatly confirming the effectiveness of MMA. Our project is released at https://luogen1996.github.io/lavin.

## 1 Introduction

In recent years, large language models (LLMs) [3, 37, 5, 52, 38] have continuously pushed the upper limit of natural language understanding with ever increasing parameter sizes and pre-training data scales. The introduction of *instruction tuning* [30, 31, 35] also enables LLMs to engage in human-like conversations and handle various natural language processing (NLP) tasks [29, 44, 45], approaching artificial general intelligence, *e.g.*, GPT-3.5 [33]. The next milestone is often regarded to extend these LLMs with multimodal capabilities, *e.g.*, vision-language (VL) learning, making LLMs applicable to more real-world application scenarios. Such a target has been recently realized by GPT-4 [34], which is likely to adopt a large-scale vision-language corpus to directly train a multimodal GPT.

---

*corresponding author

37th Conference on Neural Information Processing Systems (NeurIPS 2023).

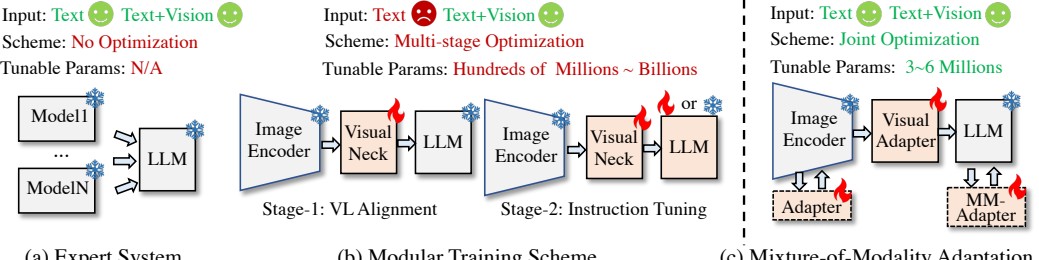

Figure 1: Comparison of different multimodal adaptation schemes for LLMs. In the expert system, LLMs play a role of controller, while the ensemble of LLM and vision models is expensive in terms of computation and storage overhead. The modular training regime (b) requires an additional large neck branch and another large-scale pre-training for cross-modal alignment, which is inefficient in training and performs worse in previous NLP tasks. In contrast, the proposed Mixture-of-Modality Adaption (MMA) (c) is an end-to-end optimization scheme, which is cheap in training and superior in the automatic shift between text-only and image-text instructions.

However, the training regime of GPT-4 [34] is prohibitively expensive, and recent endeavors [49, 50, 41, 17, 21, 1, 8, 56, 4] are still keen to efficient VL adaptions of LLMs. As shown in Fig. 1, the existing multimodal solutions for LLMs can be roughly divided into two main categories, *i.e.,* the *expert system* and the *modular training* ones, respectively. In the expert system solution [49, 50, 41], LLMs usually serve as a manager to interpret different natural language instructions, and then call the corresponding vision models to handle the input image, *e.g.*, image captioning [18, 27], visual question answering [55, 28] or text-to-image generation [39]. The advantage of this solution is that it does not require the re-training of LLMs and can make full use of existing vision models. However, the ensemble of LLMs and various vision models still exhibits significant redundancy in terms of computation and parameters, leading to excessive memory footprints. Meanwhile, the joint optimization of LLMs and vision models is still an obstacle.

In this case, increasing attention has been paid to the modular training of LLMs [17, 21, 56, 15, 56]. As illustrated in Fig. 1, this paradigm often requires LLMs to deploy an additional neck branch to connect the visual encoders, and then performs another pre-training on numerous image-text pairs for cross-modal alignment. Afterwards, the neck branch and LLM are jointly tuned via VL instructions. Despite the effectiveness, the required VL pre-training is still expensive for a quick adaptation of LLMs. For instance, the pre-training of BLIP2 [17] consumes more than 100 GPU hours on 129 millions of image-text pairs. In addition, this paradigm often requires to update most parameters of LLM, limiting the efficiency of VL instruction tuning. For example, LLaVA-13B [21] fully fine-tunes the entire LLM during VL instruction tuning, resulting in significant increases in training time and intermediate storage overhead[2]. More importantly, these fine-tune schemes will inevitably undermine the NLP capabilities of LLMs due to the drastic changes in their parameter spaces. For instance, the existing multimodal LLMs, such as BLIP2 [17] and miniGPT4 [56], do not support text-only instructions, greatly hindering their applications.

In this paper, we propose a novel and efficient solution for vision-language instruction tuning, termed *Mixture-of-Modality Adaptation* (MMA). Different from existing *modular training* scheme [17, 21], MMA is an end-to-end optimization regime. By connecting the image encoder and LLM with lightweight adapters, MMA can jointly optimize the entire multimodal LLM via a small number of parameters, saving more than thousands times of storage overhead compared with existing solutions [21, 56, 17]. To obtain a quick shift between text-only and image-text instructions, MMA equips the inserted adapters with a routing scheme, which can dynamically choose the suitable adaptation path for the inputs of different modalities, thereby well preserving the NLP capability of LLMs. To validate MMA, we apply it to a recently proposed LLM called LLaMA [43], and term this new *large vision-language instructed* model as LaVIN. With the help of MMA, LaVIN can achieve cheap and quick adaptations on VL tasks without the requirement of another large-scale pre-training.

To validate LaVIN, we first conduct quantitative experiments on ScienceQA [24]. Experimental results show that LaVIN can achieve on-par performance with the advanced multimodal LLMs, *e.g.,* LLaVA [21], while reducing up to 71.4% training time and 99.9% storage costs. Notably, fine-tuning

---

[2]The checkpoints are often stored during training, and each of them takes up 26GB for storage.

LaVIN on ScienceQA only takes **1.4 hours** with 8 A100 GPUs, and the updated parameters are only **3.8M**. In addition, we also extend LaVIN to a multimodal chatbot via tuning on 52$k$ text-only instructions [42] and 152$k$ text-image pairs [21]. The qualitative comparisons show that LaVIN can accurately execute various types of human instructions, *e.g.,* coding, math and image captioning, while yielding superior vision-language understanding than existing multimodal chatbots [56, 17, 50].

In summary, our contributions are three folds:

- We present a novel and efficient solution for vision-language instruction tuning, namely Mixture-of-Modality Adaptation (MMA), which does not require the expensive VL pretraining and can maintain the NLP capabilities of LLMs.

- Based on MMA, we propose a new multimodal LLM, namely LaVIN. Experimental results show the superior efficiency and competitive performance of LaVIN against existing multimodal LLMs, and also confirm its great potential as a general-purpose chatbot.

- We release the source code and pre-trained checkpoints associated with this paper. We believe that our project can well facilitate the development of multimodal LLM.

## 2 Related Work

### 2.1 Parameter-Efficient Transfer Learning

Since large language models have ever-increasing parameter sizes, parameter-efficient transfer learning (PETL) [13, 19, 25, 14, 22, 12] has gained increasing attention to reduce training and storage overhead of LLMs. PETL aims to insert or fine-tune a small number of parameters into LLMs, thereby achieving the adaption on downstream tasks. In early efforts [13, 12], a small MLP network, known as Adapter [13], is inserted into LLMs to project their hidden features to the semantic spaces of downstream tasks. Based on Adapter, numerous PETL methods [19, 46, 25, 14, 22, 12] have been proposed to further enhance adaptation capabilities [19, 46, 25, 22, 12] and inference speed [14]. Among them, AdaMix [46] is a method relatively close to our MMA, which also includes a set of candidate adapters for downstream task routing. However, AdaMix is static and task-dependent, of which routing path is fixed after training. In contrast, our MMA is a dynamic method based on the input modality embeddings. Moreover, AdaMix is still an unimodal module and hard to adaptively adjust the adaptions of different modalities. Driven by the great success in NLP, PETL has also achieved significant progresses in large vision models [26, 2, 54], *e.g.,* ViT [7] and CLIP [36]. Despite the effectiveness, PETL for multimodal LLMs still lacks explorations. A very recent PETL method [51] is proposed for multimodal LLMs , but its performance still lags behind full fine-tuning.

### 2.2 Multimodal Instruction-following LLMs

Instruction tuning [30, 31, 35, 47, 48] aims to fine-tune LLMs on natural language corpus describing diverse NLP tasks. This simple and effective method has been successfully applied to various well-known LLMs, such as InstructGPT [35] and FLAN-T5 [6], greatly improving their performance and generalization ability. Motivated by this success, numerous efforts have been devoted to constructing multimodal instruction-following LLMs. Existing works can be categorized into two groups, *e.g.,* the expert systems [49, 50, 41] and modular training ones [17, 21, 56, 15, 56], respectively. The representative expert systems, such as Visual ChatGPT [49] and MMREACT [50], employ LLMs as the controller to invoke various vision models to accomplish the VL instructions. Despite the effectiveness, this heavy system also incurs non-negligible burdens in terms of storage and computation. Recently, modular training models [17, 21, 56, 15, 56] as proposed as more efficient alternatives. Among them, Flamingo [1] is the first large-scale multimodal LLM that pre-trains on numerous image-text pairs, which demonstrates strong zero-shot ability on diverse tasks. The following works, including BLIP-2 [17], FROMAGe [16], PaLM-E [8], KOSMOS-1 [15] and LLaVA [21], not only optimize the model architecture [17, 16, 8, 15] but also improve the quality of VL instruction data [21]. Despite their effectiveness, most multimodal LLMs require expensive training costs and perform worse on text-only instructions.

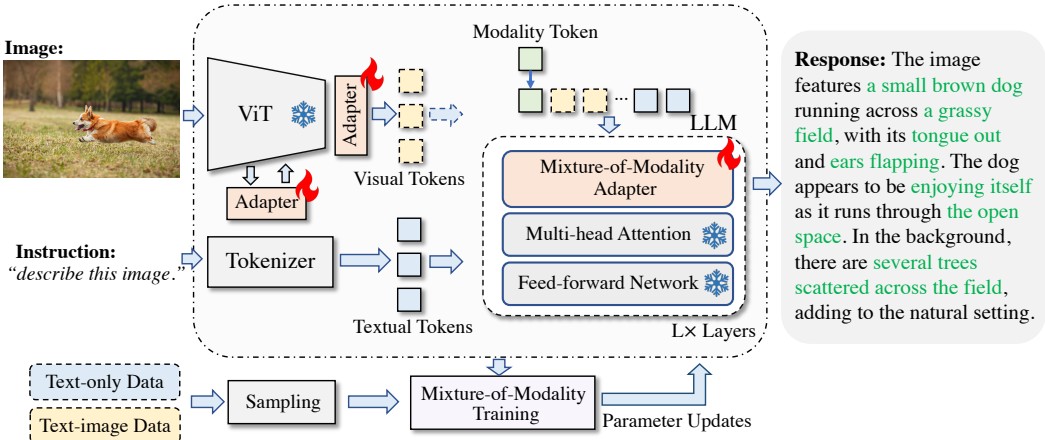

Figure 2: The overview of the Mixture-of-Modality Adaptation (MMA) and the architecture of LaVIN. In LaVIN, the novel Mixture-of-Modality Adapters are employed to process the instructions of different modalities. During instruction tuning, LaVIN is optimized by Mixture of Modality Training (MMT) in an end-to-end manner.

# 3 Method

## 3.1 Mixture-of-Modality Adaptation

In this paper, we propose a novel learning regime for the vision-language adaption of LLMs, which is called Mixture-of-Modality Adaptation (MMA). As shown in Fig. 2, MMA includes two novel designs, namely Mixture-of-Modality Adapter (MM-Adapter) and Mixture-of-Modality Training (MMT). Specifically, MM-Adapter extends LLMs with multimodal abilities via lightweight adapters, which also realizes the automatic shift between single- and multi-modal instructions. Afterwards, the entire multimodal LLM is jointly optimized via MMT, which is cheap in training time and storage.

**Mixture-of-Modality Adapter (MM-Adapter).** As shown in Fig. 2, we connect the LLM with the image encoder with a set of lightweight adaptation modules. In the image encoder, these modules can be the common adapters [13, 26]. In the LLM, unimodal adaptation modules are inferior in handling single- and multi-modal instructions simultaneously.

In particular, we first introduce a modality token $t_m \in \mathbb{R}^c$ to indicate the input modality, which is defined by

$$t_m = mE_m. \qquad (1)$$

Here, $E_m \in \mathbb{R}^{2 \times c}$ is the modality embedding. $m \in \mathbb{R}^2$ is a one-hot vector to represent the input modality. Based on the modality token $t_m$, MM-Adapter can dynamically adjust the adaptations for the input features $Z \in \mathbb{R}^{n \times c}$. In practice, $Z$ can be the single- or multi-modal features, which will be introduced in Sec 3.2. Thus, MM-Adapter can be defined by

$$Z' = Z + s \cdot router\big(f_{a_1}(Z), f_{a_2}(Z); f_w(t_m)\big). \qquad (2)$$

Here, $f_{a_1}$ and $f_{a_2}$ are RepAdapters [26] in our paper. $s$ is the scale factor, and $router(\cdot)$ is a routing function to decide the routing path of two adapters. To further reduce the parameter costs, the downsampling projection of two adapters are shared.

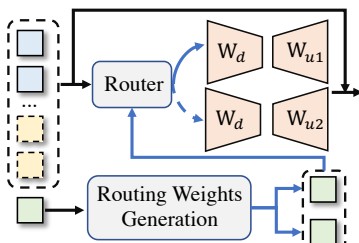

Figure 3: Illustration of the Mixture-of-Modality Adapter (MMA). MMA can dynamically select the appropriate adapter according to the input modalities.

As shown in Fig. 3, the key to realize the dynamic adaptations lies in the design of the routing function $router(\cdot)$, which is formulated as

$$router\big(f_{a_1}(Z), f_{a_2}(Z)\big) = \hat{w}_0 \cdot f_{a_1}(Z) + \hat{w}_1 \cdot f_{a_2}(Z),$$
$$\text{where} \quad \hat{w} = f_w(t_m) = \text{softmax}(\frac{t_m W_m + b_m}{\tau}). \qquad (3)$$

Here, $W_m \in \mathbb{R}^{c \times 2}$ and $b_m \in \mathbb{R}^2$ are the weight matrix and bias, respectively. $\hat{w}$ denotes the routing weights, and $\tau$ is the temperature of the softmax. Based on Eq. 2 and 3, MM-Adapter can select the best adaption path according to the modalities of input instructions. More importantly, the process of MM-Adapter only introduces a few of additional parameters, which is still efficient. In practice, MM-Adapter can be used as the unimodal adapter to improve the adaptation ability, thus we also apply it to the image encoder.

**Mixture-of-Modality Training (MMT).** Based on MM-Adapter, the target of MMT is to freeze the large image encoder and LLM, and only fine-tune the inserted adapters. In this case, the entire multimodal LLM can be jointly optimized in an end-to-end manner. Specifically, the end-to-end optimization objective can be formulated by

$$\arg\min \mathcal{L}(f_\phi(Z), R; \theta_a). \tag{4}$$

Here, $R$ and $\mathcal{L}(\cdot)$ denote the ground-truth response [24] and the objective loss function, respectively. $f_\phi$ is the LLM, and $\theta_a$ denotes the adaptation parameters. $I \in \mathbb{R}^{h \times w \times 3}$ and $T \in \mathbb{R}^l$ denote the input image and text instruction, respectively.

During training, we construct a mini training batch randomly sampled from text-only and text-image instructions. In this case, the overall training objective $\mathcal{L}$ can be defined by

$$\mathcal{L} = \sum_{i=1}^{m} \sum_{s=1}^{S+1} \log p(R_s^i | Z^i, R_{0:s-1}^i; \theta_a). \tag{5}$$

Here, $m$ denotes the batch size, and $S$ is the length of the response. After MMT, the multimodal LLM can effectively execute the input instructions of different modalities.

In our training scheme, the number of optimized parameters is still kept at a very small scale, *e.g.,* $3\sim5$M, which greatly reduces the training time and the storage cost. Compared to existing modular training paradigm, MMA does not require additional VL pre-training and can optimize the entire model end-to-end, further improving the training efficiency.

## 3.2 Large Vision-language Instructed Model

To validate MMA, we apply it to an LLM called LLaMA [43] and adopt CLIP-ViT [36] as the image encoder. Here, we term this new large vision-language instructed model as LaVIN.

Given the input image $I \in \mathbb{R}^{h \times w \times 3}$, we use the [cls] tokens from every fourth layer of ViT [7] as the visual feature, denoted as $X \in \mathbb{R}^{n \times d}$. In the image encoder, we insert the adapters before the multi-head attention modules. We represent the text instruction with word embeddings, denoted as $Y \in \mathbb{R}^{l \times c}$. Then, a simple visual adapter is used to transform the visual features to the same dimension with the LLM, which is defined by

$$X' = \sigma(X W_d + b_d) W_u + b_u. \tag{6}$$

Here, $W_d \in \mathbb{R}^{d \times d_h}$ and $W_u \in \mathbb{R}^{d_h \times c}$ denote the weight matrices, while $W_d \in \mathbb{R}^{d_h}$ and $b_u \in \mathbb{R}^c$ are the bias terms. $\sigma$ is the SwiGLU activation function [40]. In practice, $d_h$ is much smaller than $d$ and $c$, so the input of LLM can be defined by

$$Z = \begin{cases} [t_m, X', Y] & \textit{text-image}, \\ [t_m, Y] & \textit{text only}. \end{cases} \tag{7}$$

Here, $[\cdot]$ denotes the concatenation. Based on the multimodal input, LLM can predict the next token step by step, which can be formulated by

$$p_t = \prod_{s=1}^{S+1} p(R_s | Z, R_{0:s-1}; \theta_l, \theta_a) \tag{8}$$

Here, $p_t \in \mathbb{R}^m$ denotes the probabilities of the predicted word and $m$ is the length of the word embeddings. $\theta_l$ and $\theta_a$ denote the parameters of LLM and adaptation modules, respectively.

Compared with previous works [17, 56, 21], the architecture of LaVIN is much simpler and more lightweight, which is also easier to optimize. For example, the visual neck of LaVIN is 6 times smaller than that of LLaVA [21], but the performance of two models is close.

| Method | #T-Param | LLM | Subject | | | Context Modality | | | Grade | | Average |
|---|---|---|---|---|---|---|---|---|---|---|---|
| | | | NAT | SOC | LAN | TXT | IMG | NO | G1-6 | G7-12 | |
| *Zero- & few-shot methods* | | | | | | | | | | | |
| Human [24] | - | ✗ | 90.23 | 84.97 | 87.48 | 89.60 | 87.50 | 88.10 | 91.59 | 82.42 | 88.40 |
| GPT-3.5 [24] | - | ✓ | 74.64 | 69.74 | 76.00 | 74.44 | 67.28 | 77.42 | 76.80 | 68.89 | 73.97 |
| GPT-3.5 (CoT) [24] | - | ✓ | 75.44 | 70.87 | 78.09 | 74.68 | 67.43 | 79.93 | 78.23 | 69.68 | 75.17 |
| GPT-4 [34] | - | ✓ | 84.06 | 73.45 | 87.36 | 81.87 | 70.75 | 90.73 | 84.69 | 79.10 | 82.69 |
| *Representative & SoTA models* | | | | | | | | | | | |
| UnifiedQA [24] | 223M | ✗ | 71.00 | 76.04 | 78.91 | 66.42 | 66.53 | 81.81 | 77.06 | 68.82 | 74.11 |
| MM-CoT$_{Base}$ [53] | 223M | ✗ | 87.52 | 77.17 | 85.82 | 87.88 | 82.90 | 86.83 | 84.65 | 85.37 | 84.91 |
| MM-CoT$_{Large}$ [53] | 738M | ✗ | 95.91 | 82.00 | 90.82 | 95.26 | 88.80 | 92.89 | 92.44 | 90.31 | 91.68 |
| LLaVA [21] | 13B | ✓ | 90.36 | 95.95 | 88.00 | 89.49 | 88.00 | 90.66 | 90.93 | 90.90 | 90.92 |
| *Parameter-efficient methods* | | | | | | | | | | | |
| LLaMA-Adapter [51] | 1.8M | ✓ | 84.37 | 88.30 | 84.36 | 83.72 | 80.32 | 86.90 | 85.83 | 84.05 | 85.19 |
| LaVIN-7B (ours) | 3.8M | ✓ | 89.25 | 94.94 | 85.24 | 88.51 | 87.46 | 88.08 | 90.16 | 88.07 | 89.41 |
| LaVIN-13B (ours) | 5.4M | ✓ | **90.32** | 94.38 | 87.73 | **89.44** | **87.65** | 90.31 | 91.19 | 89.26 | 90.50 |
| LaVIN-13B† (ours) | 5.4M | ✓ | 89.88 | **94.49** | **89.82** | 88.95 | 87.61 | **91.85** | **91.45** | **89.72** | **90.83** |

Table 1: Comparison on ScienceQA *test* set. Question classes: NAT = natural science, SOC = social science, LAN = language science, TXT = text context, IMG = image context, NO = no context, G1-6 = grades 1-6, G7-12 = grades 7-12. † denotes that LaVIN is trained with 40 epochs. #T-Params denotes that the number of trainable parameters.

| Settings | #T-Params | NAT | SOC | LAN | TXT | IMG | NO | G1-6 | G7-12 | Avg. |
|---|---|---|---|---|---|---|---|---|---|---|
| Text Only | 1.8M | 82.86 | 82.56 | 82.28 | 81.23 | 75.81 | 86.06 | 83.26 | 81.54 | 82.65(+0.00) |
| + Vision Modality (MMT) | 2.4M | 85.97 | 90.66 | 83.55 | 84.90 | 83.59 | 86.41 | 88.14 | 83.06 | 86.32(+3.67) |
| + Joint Opt. (MMT) | 2.5M | 86.59 | 94.71 | 82.91 | 85.63 | 84.98 | 86.41 | 88.62 | 85.04 | 87.34(+4.69) |
| + Stronger Image Enc. | 2.9M | 88.01 | 94.94 | 83.64 | 87.15 | 86.81 | 87.04 | 89.87 | 85.56 | 88.33(+5.68) |
| + MM-Adapter | 3.8M | 89.25 | 94.94 | 85.24 | 88.51 | 87.46 | 88.08 | 90.16 | 88.07 | 89.41(+6.76) |
| + Larger LLM (13B) | 5.4M | 90.32 | 94.38 | 87.73 | 89.44 | 87.65 | 90.31 | 91.19 | 89.26 | 90.50(+7.85) |

Table 2: Ablation studies on ScienceQA *test* set. For the text-only baseline, we use the image caption to prompt the model. ViT-B/16 and LLaMA-7B are used as the default image encoder and LLM. "Joint Opt" denotes the joint optimization of image encoder and LLM. The Mixture-of-Modality Training (MMT) is ablated with the settings of "Vision Modality" and "Joint Opt.".

## 4 Experiments

### 4.1 Datasets and Metrics

**ScienceQA.** ScienceQA [24] is the large-scale multimodal dataset for science question answering, which covers various domains, including 3 subjects, 26 topics, 127 categories and 379 skills. ScienceQA consists of text-only and text-image examples in three splits namely *train*, *val* and *test*, with 12,726, 4,241 and 4,241 examples, respectively. We evaluate our model using average accuracy.

**Alphaca-52k & LLaVA-158k.** Alphaca-52k [42] contains 52k text-only instruction-following data generated by GPT-3.5 [3]. LLaVA-158k [21] is a large-scale text-image instruction-following dataset, where the answer is automatically generated by GPT-4 [34]. Following LLaVA [21], GPT-4 is employed to evaluate the quality of the chatbot's responses, which will assign higher scores to superior responses within a range of 1 to 10.

### 4.2 Implementation Details

We employ the ViT-L/14 [7] of the pre-trained CLIP [36] as the image encoder. The visual features consist of six `[cls]` tokens extracted from every fourth layer of ViT-L/14. For LLM, LLaMA-7B [43] and LLaMA-13B [43] are used. The default dimension of the visual neck is set to 128. The dimension of MM-Adapter is 8, and the temperature is set to 10 for LaVIN-7B and 5 for LaVIN-13B. For text-only baseline, the image encoder is removed, and MM-Adapter is replaced with RepAdapter [26]. We adopt AdamW [23] as the optimizer, and train the model for 20 epochs with a cosine decay learning rate schedule. The batch size, learning rate and weight decay are set to 32, 9e-3 and 0.02, respectively. During the generation stage, the decoding uses *top-p* sampling with a temperature of 0.1 and a *top-p* value of 0.75, respectively. For the experiments of multimodal chatbot, all hyperparameters remain the same, except for the training epochs, which are reduced to 15.

## 4.3 Experimental Results

### 4.3.1 Quantitative Experiments

**Results on ScienceQA.** In Tab. 1, We first compare LaVIN with the state-of-the-art methods on ScienceQA. From this table, the first observation is that the few-shot LLMs, such as GPT-4, still perform worse than human, suggesting the great challenge of ScienceQA. In contrast, existing supervised methods [21, 51, 53] yield better results. In particular, MM-CoT$_{Large}$ [53] achieves the best performance, *e.g.,* 91.68. However, MM-CoT mainly focuses on the multi-modal chain-of-thought for language models, of which contribution is orthogonal to our approach. In particular, LLaVA [21] is an end-to-end multimodal LLM, which is more close to our work.

The results show that LLaVA remains competitive performance against MM-CoT$_{Large}$[53], especially in the category of SOC. Despite the effectiveness, its number of trainable parameters is still large, leading to higher training overhead. LLaMA-Adapter [51] adopts a parameter-efficient scheme to reduce the training overhead, but its performance still greatly lags behind LLaVA. Compared to these approaches, LaVIN achieves the better trade-offs between performance and training efficiency. For exam-

| Methods | #T-Params | Accuracy |
|---|---|---|
| LLaVA [21] | 13B | 85.81 |
| LLaMA-Adapter [51] | 1.8M | 85.19 |
| LaVIN-7B | 3.8M | 89.41 (+4.22) |
| LaVIN-13B | 5.4M | **90.83** (+5.02) |

Table 3: Results of LaVIN and existing multimodal LLMs without the pre-training stage. We report the average accuracy on ScienceQA *test* set.

ple, LaVIN-7B consumes a similar scale of trainable parameters as LLaMA-Adapter [51], while outperforming it by +4.22 gains. When scaling up to 13B, LaVIN can obtain more significant performance gains, *i.e.,* +5.64. Compared to LLaVA, LaVIN-13B also achieves comparable performance and even performs better in some question classes, *e.g.,* LAN and NO. Considering the much lower training costs than LLaVA, such competitive performance greatly confirms the efficiency and designs of LaVIN.

In Tab. 3, we compare LaVIN with existing methods without VL pre-training. From this table, we observe that both LLaVA [21] and LLaMA-Adapter achieve the similar performance, *i.e.,* 85.81 *vs.* 85.19. In particular, LLaVA [21] and LLaMA-Adapter [51] freeze the image backbone, and the entire multimodal LLM is not jointly optimized, which hinders the learning of visual content. Moreover, the adaptation module in LLaMA-Adapter does not consider

| Methods | PT Data | #T-Params | BLEU-4 | CIDEr |
|---|---|---|---|---|
| ClipCap [32] | 0 | - | 33.5 | 113.1 |
| LLaMA-Adapter V2 [11] | 0 | 14M | 36.2 | 122.2 |
| BLIP [18] | 14M | 583M | 40.4 | 136.7 |
| BLIP-2 [17] | 129M | 188M | 43.7 | 145.3 |
| LaVIN (ours) | 0 | 5.4M | 36.4 | 126.9 |
| LaVIN (ours) | 0.6M | 5.4M | 37.8 | 131.7 |

Table 4: Fine-tuning results of LaVIN and existing multimodal LLMs on COCO captioning. We report performance on *Karpathy* test split.

the modality gap in the input instructions, greatly limiting its performance upper bound. In contrast, with the help of MMA, LaVIN significantly outperforms these approaches, *e.g.,* +5.02 gains over LLaVA. These results validate the proposed MMA towards the effective and efficient VL adaption, and confirm the designs of LaVIN.

**Results on COCO Captioning.** In Tab 4, we compare LaVIN with existing methods on the task of image captioning. From these results, we can still observe the competitive performance of LaVIN. As a parameter-efficient tuning method, LaVIN outperforms LLaMA-Adapter v2 [11] by a large margin, *e.g.,* up to +9.5 of CIDEr. Compared with large-scale pre-training models, *e.g.,* BLIP and BLIP-2, the performance of LaVIN is still comparable, while the expenditure is much cheaper. For instance, with only 0.6M pre-training data and 5.4M updated parameters, LAVIN can achieve 131.7 CIDEr on COCO Captioning. Notably, our tuning only takes 4 GPU hours on 8 A100s, while BLIP-2 requires more than 300 GPU hours on 16 A100s. These results further validate the effectiveness and training efficiency of MMA and LaVIN.

**Zero-shot evaluation on NLP and multimodal benchmarks.** In Tab. 5, we evaluate the zero-shot ability of LaVIN and existing methods on TruthfulQA [20] and MME [10]. On TruthfulQA [20], we observe that the zero-shot performance of existing multimodal LLMs is obviously inferior to the original LLaMA. In stark contrast, LaVIN can further improve the performance by +9.2%

than LLaMA-Base [43] through its mixture-of-modality adaptation. On MME [10], a challenging benchmark for multimodal evaluation, LaVIN still demonstrates competitive performance against existing multimodal LLMs. Expect for BLIP-2 [17], which is pre-trained on numerous data, the other methods perform similarly to or worse than LaVIN, *e.g.,* 866.5 of MiniGPT-4 *vs.* 963.6 of LaVIN on MME-C. These results confirm the strong generalization ability of LaVIN, and also validate that the NLP capabilities are well preserved by MMA during VL instruction tuning.

**Ablation study.** To gain deep insights into MMA and LaVIN, we conduct comprehensive ablation studies in Tab. 2. From this table, we can see that each design of MMA and LaVIN greatly contributes to the final performance. As shown in Tab. 2, the mixture-of-modality training (MMT) brings the most significant gains, *e.g.,* +4.69. In MMT, the joint training with the vision modality provides up to +3.67 performance gains for LaVIN. With the joint optimization of the image encoder and LLM, the performance of LaVIN further boosts from 86.32 to 87.34, suggesting the significance of the joint optimization for multimodal LLMs. With the help of MMT, LaVIN already surpasses the existing parameter-efficient method, *i.e.,* LLaMA-Adapter. Additionally, the stronger image encoder,

| Methods | TruthfulQA | MME-C | MME-P |
|---|---|---|---|
| LLaMA-Base [43] | 38.7 | - | - |
| LLaMA-Adapter V2 [11] | 24.4 | 972.6 | 248.9 |
| LLaVA [21] | 16.4 | 502.8 | 214.6 |
| BLIP-2 [17] | - | 1293.8 | 290.0 |
| MiniGPT-4 [56] | - | 866.5 | 292.1 |
| LaVIN (ours) | 47.9 | 963.6 | 249.6 |

Table 5: Zero-shot results on NLP and multimodal benchmarks. "*Mc1_targets*" setup is used on TruthfulQA [20]. "MME-C" and "MME-P" denote the splits of *Cognition* and *Perception* on MME benchmark [10], respectively.

*i.e.,* ViT-L/14, also improves the average accuracy by 0.99. An interesting observation is that a better image encoder provides noticeable performance gains for both image-based and text-based questions. When adopting MM-Adapter to LaVIN, we observe +1.08 gains on average accuracy. Such an improvement only requires extra 0.9M parameters, which is very lightweight. Meanwhile, the performance of LaVIN is significantly improved by MM-Adapter on more challenging metrics like G7-12, *i.e.,* +2.51. After scaling up LLM to 13B, the performance of LaVIN is further improved by + 1.09. Overall, these ablations well validate the significance of MMA in adapting multimodal LLM, and also confirm the effectiveness of LaVIN.

**Comparison of training efficiency.** In Tab. 6, we compare the training expenditures of LaVIN, LLaVA [21] and BLIP2 [17]. The first observation is that the pre-training cost of BLIP2 is actually expensive, which requires more than 200 hours. Meanwhile, LLaVA cannot be trained on common machines with the default training settings[3]. Thus, it requires some GPU memory-saving techniques [9] to avoid *out of memory* (OOM). However, its training time and storage requirement are still significant. For example, it still takes up to 26GB space to store the updated parameters of the LLM. In contrast, LaVIN demonstrates superior training efficiency with the help of MMA. Compared to LLaVA,

| Methods | #T-Params | Memory | Time | #Storage |
|---|---|---|---|---|
| BLIP2 [17] | 188M | - | >200 hours | - |
| LLaVA [21] | 13B | OOM | N/A | N/A |
| LLaVA‡ [21] | 13B | 36.8G | 7 hours | 26GB |
| LaVIN-7B | 3.8M | 33.9G | 1.4 hours | 15M |
| LaVIN-13B | 5.4M | 55.9G | 2 hours | 20M |

Table 6: Training costs of LaVIN and existing multimodal LLMs on ScienceQA. ‡ denotes that GPU memory-saving techniques are used. "OOM" denotes out of GPU memory. All results are evaluated on 8 A100 GPUs.

LaVIN-7B and LaVIN-13B reduce about 80% and 71.4% training time, respectively. In terms of GPU memory and storage cost, our approach can save more than 40% GPU memory and 99.9% disk storage. Overall, these results greatly confirm the training efficiency of MMA.

### 4.3.2 Qualitative Experiments

**Examples of different instruction-following tasks.** In Fig 4, we compare LaVIN with existing methods [51, 21] on single- and multi-modal instruction-following tasks, *e.g.,* math, coding and image captioning. Compared to LLaVA [21] and LLaMA-Adapter [51], LaVIN achieves overall better responses across multiple tasks. In Fig.4 (a), LaVIN correctly answers the math problem with a result of 28.8, whereas LLaMA-Adapter [43] provides an incorrect answer. In example (d), LaVIN generates accurate code for the request of "*print prime numbers up to 100*". In contrast, the

---

[3]https://github.com/haotian-liu/LLaVA

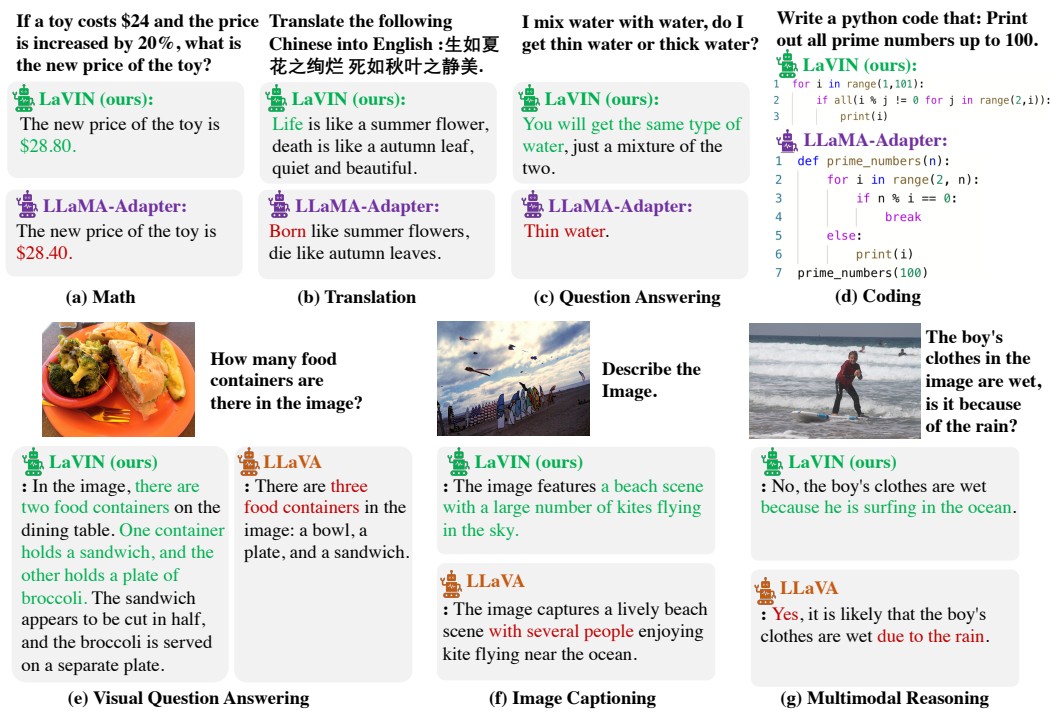

Figure 4: Comparison between LaVIN-13B and existing methods on single- and multi-modal instructions. The noteworthy aspects of the responses are highlighted in green, whereas the illogical portions are marked in red. More tasks and examples are given in appendix.

code written by LLaMA-Adapter is to check prime numbers, which does not produce any output during execution. Meanwhile, LaVIN presents a clear and concise coding behavior, acting more like a professional programmer. In Fig 4 (e)-(g), LaVIN demonstrates remarkable visual reasoning ability in accomplishing various multimodal tasks. In Fig.4 (e), LaVIN accurately answers the complex questions about the number of food containers in the image and provides a detailed description about the complex scene. The same observation can also be witnessed in Fig.4 (g), where LaVIN infers a correct reason for the wetness of the boy's clothes. Overall, these examples show the superior reasoning ability of LaVIN in executing single- and multi-modal instructions, while also confirming the significance of MMA in adapting LLMs to multi-modal tasks.

**Examples of multimodal dialogue** In Fig. 5, we compare LaVIN with existing multimodal LLMs in multi-turn conversations, and use GPT4 [34] to evaluate the quality of their responses. From the results, we can see that LaVIN has higher GPT4 scores among all compared models, suggesting superior ability in multimodal dialogue. Meanwhile, we also observe different response styles of these multimodal LLMs. In particular, BLIP2 [17] tends to produce brief responses, which lack detailed explanations. In contrast, the responses of MiniGPT4 [56] are the longest among all models, but their content is often redundant and repetitive. Compared to them, LaVIN and LLaVA [21] can generate more accurate responses. Particularly, LaVIN performs better than the other methods, mainly due to its more logical and detailed descriptions. As illustrated in the first question, LaVIN not only provides the correct answer, but also explains the reason behind it. In the second question, LaVIN and LLaVA are required to judge whether the man will get wet, and LaVIN answers "yes" while LLaVA considers "no". It can be seen that the reason of LaVIN is more comprehensive, logical and persuasive than LLaVA, which considers the situation of "*the overhand may not provide the complete protection*". Overall, these examples confirm that MMA equips LLMs with excellent multi-modal ability, requiring no pre-training on large-scale image-text data.

## 5 Limitations and Broader Impact

We observe two primary limitations of LaVIN. Firstly, LaVIN may generate incorrect or fabricate responses, similar to existing multimodal LLMs. Secondly, LaVIN can not identify extremely fine-

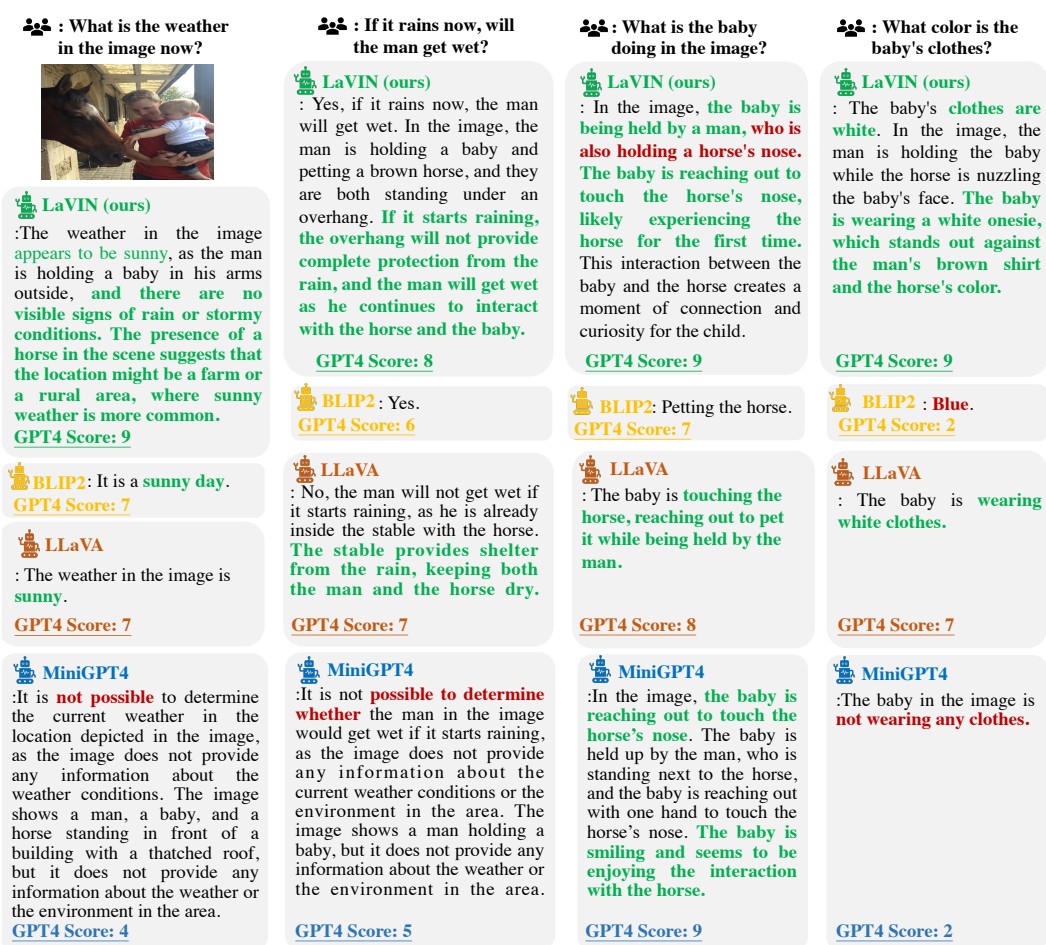

Figure 5: Comparison of LaVIN-13B and existing multimodal LLMs in multi-turn conversations. GPT-4 assigns a score ranging from 1 to 10 to evaluate the quality of a response, with a higher score indicating superior performance. The noteworthy aspects of the responses are highlighted in **green**, whereas the illogical portions are marked in **red**.

grained visual content, such as text characters. We believe that the recognition ability of LaVIN still has a large room to improve, which will be left in our future work.

# 6   Conclusions

In this paper, we propose a novel and affordable solution for vision-language instruction tuning, namely Mixture-of-Modality Adaptation (MMA). Particularly, MMA is an end-to-end optimization regime, which connects the image encoder and LLM via lightweight adapters. With the help of MMA, the entire multimodal LLM can be jointly optimized via a small number of parameters, greatly reducing the training costs. Meanwhile, we also propose a novel routing algorithm in MMA, which can help the model automatically shifts the reasoning paths for single- and multi-modal instructions. Based on MMA, we develop a large vision-language instructed model called LaVIN, which demonstrates a superior reasoning ability than existing multimodal LLMs in various instruction-following tasks.

**Acknowledgements.**   This work was supported by National Key R&D Program of China (No.2022ZD0118201) , the National Science Fund for Distinguished Young Scholars (No.62025603), the National Natural Science Foundation of China (No. U21B2037, No. U22B2051, No. 62176222, No. 62176223, No. 62176226, No. 62072386, No. 62072387, No. 62072389, No. 62002305 and No. 62272401), the Natural Science Foundation of Fujian Province of China (No.2021J01002, No.2022J06001), and the China Fundamental Research Funds for the Central Universities (Grant No. 20720220068). We also thank Dr. Mingbao Lin for his valuable suggestions.

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
