# Appendix

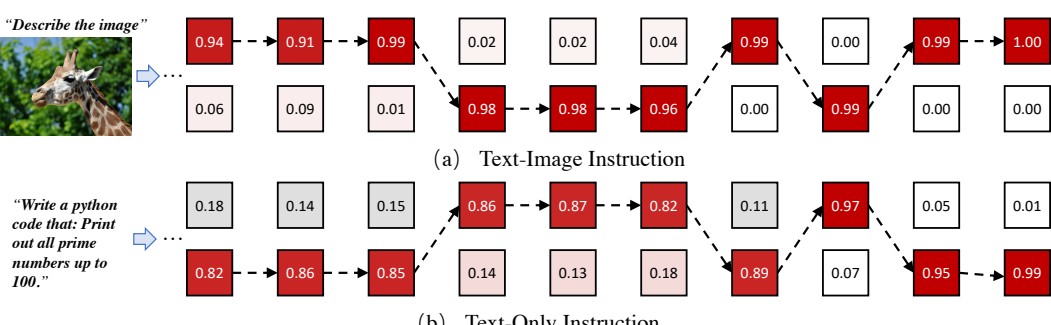

Figure 1: Visualization of the dynamic inference paths between two adapters in the last 10 layers of LaVIN-7B. The values in the box denote the routing weights. Given input instructions of different modalities, LaVIN can dynamically execute different inference paths based on the input modality embeddings.

## A More Visualizations

### A.1 Inference Path of LaVIN

In Fig. 1, we visualize the routing weights of LaVIN for text-only and text-image instruction inputs. From the figure, the first observation is that MM-Adapter effectively decouples the inference of different modalities into two set of adapters. As shown in Fig. 1, the inference path for text-image instruction input significantly differs from that of text-only instruction. Meanwhile, the learned routing weights are also very sharp, *i.e.,* close to 0 or 1, suggesting that the model is very confident to its decision. From these two observations, we can find that the text-only and text-image instruction inputs actually have different requirements for their adaptations, thus common unimodal adapters are usually suboptimal to simultaneously adapt two modalities. This visualizations also confirm the effectiveness of MM-Adapter.

### A.2 More Visualization Examples

To gain deeper insights into LaVIN, we provide more visualizations in Fig. 2, 3, 4 and 5. In Fig. 2, we compare LaVIN and existing methods on more instruction-following tasks. In Fig 3 and 4, we compare LaVIN, BLIP2, LLaVA and MiniGPT4 on more examples of multimodal dialogues. From Fig. 2, 3 and 4, we observe that LaVIN demonstrate better reasoning ability than existing methods on text-only and text-image instructions, suggesting the effectiveness of our mixture-of-modality adaptation. In Fig. 5, we provide typical failure cases of LaVIN. From it we can see that LaVIN can not correctly translate sentences with complex logic and structure. Meanwhile, LaVIN will also incorrectly describe the fine-grained information in an image. We believe that these mistakes will be addresses by scaling up the model and improving the visual representations, which will be explored in our future work.

Submitted to 37th Conference on Neural Information Processing Systems (NeurIPS 2023). Do not distribute.

**The dead don't die again, so why not let the dead go to war?**

**LaVIN (ours):**

The idea of letting the dead go to war is a controversial one, as it raises ethical questions about the morality of using the dead to fight a war. Additionally, it could lead to a situation where the dead are used as a weapon, which could have devastating consequences.

**LLaMA-Adapter:**

The dead don't die again, so why not let the dead go to war? This is a question that has been asked for centuries. The dead have been used in warfare since the beginning of time, and it is believed that the dead can be used to protect the living. The dead can be used as a source of power, as they are believed to have supernatural abilities. The dead can also be used as a distraction, as they can be used to draw the enemy's attention away from the living. The dead can also be used as a source of strength, as they can be used to motivate

**Summarize the following text in one sentence: During the second quarter, our company's revenue grew by 15% year-over-year, reaching $50 million. Our net income also improved, increasing by 20% to $10 million. We achieved this growth through successful product innovation and expanding our customer base in key markets. However, we faced increased competition and rising costs, which impacted our gross profit margin, resulting in a decline to 32%. We are implementing cost-saving measures to improve profitability in the coming quarters.**

**LaVIN (ours):**

Our company experienced 15% revenue growth and 20% net income growth in the second quarter, but rising costs and increased competition led to a decline in gross profit margin to 32%. We are implementing cost-saving measures to improve profitability in the coming quarters.

**LLaMA-Adapter:**

Our company's revenue and net income increased in the second quarter, but our gross profit margin declined due to increased competition and rising costs. We are implementing cost-saving measures to improve profitability in the coming quarters.

**Explain the following code**

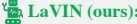

```
01.  sum=0
02.  for i in range(1,101):
03.      if i%2!=0:
04.          sum+=i
05.  print(sum)
```

**LaVIN (ours):**

The code above creates a variable 'sum' which is initialized to 0. Then, it loops through the numbers from 1 to 101, and if the number is not divisible by 2, it adds the number to the sum. At the end, the value of 'sum' is printed.

**LLaMA-Adapter:**

The code above calculates the sum of all even numbers from 1 to 101. It starts by setting the variable sum to 0. Then, it iterates through the range of numbers from 1 to 101, checking if the number is even. If it is, it adds the number to the sum. Finally, it prints the sum.

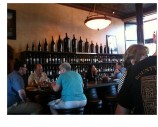

**Describe the Image.**

**LaVIN (ours)**

: The image depicts a lively restaurant scene with a group of people sitting at a bar, enjoying their time together.

**LLaVA**

: The image depicts a group of people sitting at a bar, enjoying their time together.

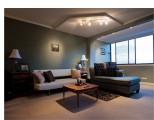

**Describe the Image.**

**LaVIN (ours)**

: The image features a cozy living room with a large couch situated in the center of the room.

**LLaVA**

: The image displays a living room with a couch, a coffee table, a lamp, and a window. The room is well-lit and clean.

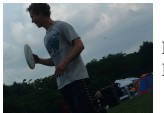

**Describe the Image.**

**LaVIN (ours)**

: The image features a man standing in a grassy area, holding a white frisbee in his hand.

**LLaVA**

: The image features a man standing in a park, holding a white frisbee in his right hand.

Figure 2: More examples of LaVIN on different instruction-following tasks.

**👥 : What is the weather in the image now?**

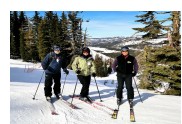

**🤖 LaVIN (ours)**
:The weather in the image is sunny, as the three people are standing on a snow-covered slope with the sun shining brightly in the background. This suggests that it is a clear and pleasant day, ideal for skiing and enjoying the outdoors.

**GPT4 Score: 9**

**🤖 BLIP2**: it is snowing.

**GPT4 Score: 4**

**🤖 LLaVA**
: The weather in the image is sunny.

**GPT4 Score: 2**

**🤖 MiniGPT4**
:I'm sorry, but I don't have access to current weather information for the location in the image. The image appears to be of a group of people standing on a snowy slope, but I don't have any information about the current weather conditions in that location.

**GPT4 Score: 8**

---

**👥 : What activities are they most likely engaged in?**

**🤖 LaVIN (ours)**
: The three people in the image are most likely engaged in skiing activities. They are standing on a snow-covered slope, wearing ski gear, and holding ski poles. This suggests that they are either preparing to ski down the slope or have just finished skiing and are taking a break to pose for a picture. The presence of skis and ski poles indicates that they are participating in a winter sport, which is typically associated with skiing.

**GPT4 Score: 9**

**🤖 BLIP2** : Skiing.

**GPT4 Score: 3**

**🤖 LLaVA**
: The three people in the image are most likely engaged in skiing, as they are standing on a snow-covered slope with skis on. They might be taking a break, posing for a photo, or preparing to continue skiing down the slope.

**GPT4 Score: 8**

**🤖 MiniGPT4**
:It is not possible for me to accurately determine the activities that the people in the image are most likely engaged in. The image appears to be of a group of people standing on a snowy slope, but I don't have any information about their activities or the context of the image. It is possible that they are engaged in skiing or other winter sports, but it is not possible for me to confirm this based on the information provided.

**GPT4 Score: 6**

---

**👥 : What is the most likely temperature in degrees Celsius in the image?**

**🤖 LaVIN (ours)**
:The most likely temperature in degrees Celsius in the image is around -10 degrees Celsius, as the three people are standing on a snow-covered slope, which is typically associated with cold weather conditions. The presence of snow and the fact that they are wearing skiing gear further supports the assumption that the temperature is below freezing.

**GPT4 Score: 9**

**🤖 BLIP2** : -5 to -10 degrees Celsius.

**GPT4 Score: 7**

**🤖 LLaVA**
: While I cannot provide the exact temperature, the image shows a sunny day with snow on the ground, which suggests that the temperature is likely to be around or slightly below the freezing point of water (0° C or 32° F). This is the ideal temperature range for skiing, as it provides a balance between the snow being sufficiently cold to remain on the ground and not too cold to make skiing uncomfortable or dangerous.

**GPT4 Score: 8**

**🤖 MiniGPT4**
:It is not possible for me to accurately determine the temperature in degrees Celsius in the image. The image does not provide enough information for me to make an informed guess about the temperature. The temperature can be affected by a variety of factors, including the time of day, the location, and the weather conditions. Without more information, it is not possible for me to accurately determine the temperature in the image.

**GPT4 Score: 5**

---

**👥 : What emotions do they convey?**

**🤖 LaVIN (ours)**
: The three people standing on the snow-covered slope in their ski gear convey a sense of happiness and excitement. They are posing for a picture, which suggests that they are enjoying their time skiing and want to capture the moment. The presence of skis and snowboards in the image further supports the idea that they are engaged in a winter sport and are enthusiastic about it.

**GPT4 Score: 9**

**🤖 BLIP2** : they are happy to pose for a picture.

**GPT4 Score: 6**

**🤖 LLaVA**
: The three people in the image convey a sense of fun, enjoyment, and camaraderie. They are smiling and posing together, which suggests that they are having a good time skiing and sharing the experience with each other. This positive atmosphere and the beautiful winter scenery create a visually appealing and uplifting image.

**GPT4 Score: 9**

**🤖 MiniGPT4**
:It is not possible for me to accurately determine the emotions that the people in the image are conveying. The image does not provide enough information for me to make an informed guess about the emotions of the people in the image. It is possible that the people in the image are feeling happy, excited, or focused, but it is not possible for me to accurately determine their emotions based on the information provided.

**GPT4 Score:8**

Figure 3: More examples of LaVIN on multimodal dialogues.

👥 : What colors does the food in the image contain?

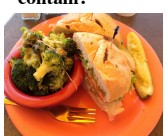

🤖 **LaVIN (ours)**
:The food in the image contains a variety of colors, including green, yellow, and red. The green color comes from the broccoli, yellow from the sandwich, and red from the sauce or dressing on the sandwich. The presence of these different colors adds visual appeal to the meal, making it more appetizing and enjoyable to look at.
**GPT4 Score: 8**

🤖 **BLIP2**: red, orange, and green.
**GPT4 Score: 4**

🤖 **LLaVA**
: The food in the image contains the colors green (from the broccoli) and yellow (from the mac and cheese).
**GPT4 Score: 6**

🤖 **MiniGPT4**
:The sandwich on the plate appears to be made with bread, lettuce, tomato, and cheese. The broccoli on the side appears to be cooked and seasoned. The fork is made of metal and has a wooden handle.
**GPT4 Score: 5**

👥 : How many kinds of food are there in the image?

🤖 **LaVIN (ours)**
: In the image, there are two kinds of food: a sandwich and broccoli. The sandwich is placed on a red plate, and the broccoli is served on a separate plate. The sandwich appears to be a vegetarian option, as it is described as a "vegetable sandwich." The presence of a fork and a knife on the table suggests that the meal is ready to be enjoyed.
**GPT4 Score: 9**

🤖 **BLIP2** : 2.
**GPT4 Score: 6**

🤖 **LLaVA**
: There are three kinds of food in the image: a sandwich, broccoli, and mac and cheese.
**GPT4 Score: 6**

🤖 **MiniGPT4**
:There are two kinds of food in the image: the sandwich and the broccoli. The sandwich is made with bread, lettuce, tomato, and cheese, and is the main food item being served. The broccoli is a side dish, and is served in a separate container on the side of the plate.The sandwich and broccoli are both types of vegetables, but they are different varieties. The sandwich is a type of sandwich, and the broccoli is a type of broccoli. The sandwich is made with bread, lettuce, tomato, and cheese, while the broccoli is simply cooked broccoli. The sandwich is the main food item being served, while the broccoli is a side dish to complement the sandwich.
**GPT4 Score: 7**

👥 :What objects are the main focus of the image?

🤖 **LaVIN (ours)**
:The main focus of the image is a plate with a sandwich and a side of broccoli. The sandwich is cut into smaller pieces, and the broccoli is served on a separate plate. The arrangement of these food items on the table creates an appetizing presentation, showcasing a balanced meal.
**GPT4 Score: 9**

🤖 **BLIP2** : a sandwich, broccoli, and a fork.
**GPT4 Score: 5**

🤖 **LLaVA**
: The main focus of the image is the plate of food, which includes a sandwich, broccoli, and mac and cheese.
**GPT4 Score: 6**

🤖 **MiniGPT4**
:The main focus of the image is the sandwich on the orange plate. The sandwich is made with bread, lettuce, tomato, and cheese, and is the most prominent object in the image. The broccoli on the side is also visible, but it is not as prominent as the sandwich. The fork is also visible, but it is a minor object in the image compared to the sandwich and broccoli.The sandwich is the main focus of the image because it is the largest and most prominent object in the image, and it is the main item being served. The broccoli is also visible, but it is a secondary object in the image, serving as a side dish to complement the sandwich. The fork is a minor object, serving as a tool to eat the food.
**GPT4 Score: 8**

👥 : Are there vegetables in the image?

🤖 **LaVIN (ours)**
: Yes, there are vegetables in the image. A broccoli piece can be seen on the plate, along with the sandwich and other food items.
**GPT4 Score: 9**

🤖 **BLIP2** : yes.
**GPT4 Score: 3**

🤖 **LLaVA**
: Yes, there are vegetables in the image, including broccoli.
**GPT4 Score: 8**

🤖 **MiniGPT4**
:Yes, there are vegetables in the image. The sandwich contains lettuce and tomato, and there is a side dish of cooked broccoli on the plate.
**GPT4 Score: 9**

Figure 4: More examples of LaVIN on multimodal dialogues.

**Translate the following English into Chinese: If you think he is a good man, think again.**

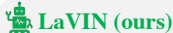 **LaVIN (ours):**
如果你觉得他是一个好人，再觉得一下。

**How do I prove I was born if I lost my birth certificate?**

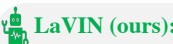 **LaVIN (ours):**

If you have lost your birth certificate, you can prove your birth by obtaining a certified copy of your birth record from the hospital where you were born. You can also obtain a copy of your birth record from the state where you were born. Additionally, you can obtain a copy of your birth record from the National Archives and Records Administration.

**Translate the following Chinese into English:在天愿作比翼鸟，在地愿为连理枝**

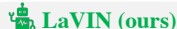 **LaVIN (ours):**
I wish to soar like a swallow in the sky, and to weave a web of connection on the ground.

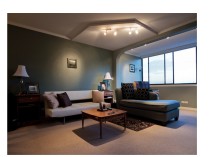 **Describe the Image.**

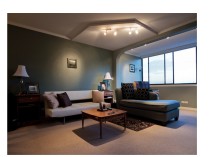 **LaVIN (ours)**

**:** The image features a cozy living room with a large couch situated in the center of the room.

**Summarize the following text in one sentence:**
**He's not perfect. You aren't either, and the two of you will never be perfect.**
**But if he can make you laugh at least once, causes you to think twice, and if he admits to being human and making mistakes, hold onto him and give him the most you can.**
**He isn't going to quote poetry, he's not thinking about you every moment, but he will give you a part of him that he knows you could break.**
**Don't hurt him, don't change him, and don't expect for more than he can give.**
**Don't analyze.**
**Smile when he makes you happy, yell when he makes you mad, and miss him when he's not there. Love hard when there is love to be had. Because perfect guys don't exist, but there's always one guy that is perfect for you.**

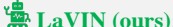 **LaVIN (ours):**

He may not be perfect, but he will give you a part of himself that he knows you could break. Give him the best of you and love him hard when there is love to be had.

Figure 5: Failure cases of LaVIN.