# OpenReview forum: "Cheap and Quick: Efficient Vision-Language Instruction Tuning for Large Language Models"
_NeurIPS.cc/2023/Conference — NeurIPS 2023 poster_

### Official Review · Reviewer_4c4u · 2023-06-12

**Soundness:** 3 good
**Presentation:** 3 good
**Contribution:** 3 good
**Rating:** 6
**Confidence:** 3

**Summary:**

This paper presents a novel method to connect vision-language instruction tuning with large language models.
To implement this, the authors introduce the adapters rather than heavy bottlenecks between vision-language input tokens and LLM.
A routing mechanism is designed to adaptively choose the right direction for different modalities of inputs.
After applying this method to the recent strong LLaMa model, the proposed method achieves efficient instruction tuning and close the performance gap with that of the original large language model.

**Strengths:**

- The paper is well-written and most parts of this paper are easy to follow.
- The proposed method achieves significant performance improvement pertaining to instruction tuning while using a very small magnitude of trainable parameters.
- The proposed method demonstrates favorable results on both single- and multi-modal instructions.

**Weaknesses:**

- My biggest concern lies in the routing mechanism.
  - Why do we need the routing between different kinds of inputs? Can we just use determined inputs to the modality-aware adapters?
  - What the role of ```s``` is is not well explained.
  - It seems the router output is simply a weighted sum of the two adapters. How should we explain this?

- It seems the authors also introduce adapters to the image encoder. This should also be explained.

- For the image input features, the authors use the ```[CLS]``` of every fourth layer of CLIP-ViT, are there any rationales behind this?

- I'm confused about Sec. 3.2. It seems that Sec. 3.2 can be cohesively organized with that of MMT. Are there more considerations about this section? If not, the writing should be better organized.

- Some typos:
  - Line 156, ```m``` and ```n``` are in reverse order;
  - Line 170, the dimension definition is not correct.




**Questions:**

see the weaknesses above

---

> ### Author Rebuttal · Authors · 2023-08-09
>
> We highly appreciate your careful review for this paper. Your beneficial feedback and valuable suggestions indeed improve our paper a lot.  Below, we response to your key concerns point by point.
>
> **Comment#1：**
>
> Why do we need the routing between different kinds of inputs? Can we just use determined inputs to the modality-aware adapters?
>
> **Response:**
>
>  Thanks for this insightful question. Compared to  our MM-adapter,  the shortcomings of modality-specific adapters  lies in two aspects:
>
> 1. **Inference flexibility.**  Modality-specific adapters require to manually set up the inference adapters according to the input modalities, while MM-adapter can automatically adjust its inference paths, which is more convenient.
> 2. **Training efficiency.** Modality-specific adapters can not be benefited from the mixture-of-modality training.  In modality-specific adapters,  each adapter is separately optimized by one kind of input, greatly reducing the training efficiency. Under the same training epoch, the performance of modality-specific adapters on ScienceQA is 83.76,  obviously inferior to 89.41 of MM-adapter.
>
> Overall, the proposed MM-adapter is still a better solution than modality-specific adapters in practice.
>
>
>
> **Comment#2：**
>
> What the role of s  is not well explained.
>
> **Response:**
>
> ''s''  is the scale factor to adjust the  numeric magnitude [12] for stable training, which is  set to 1 in our experiments. We will add more explanations in our final version.
>
>
>
> **Comment#3：**
>
> It seems the router output is simply a weighted sum of the two adapters. How should we explain this?
>
> **Response:**
>
>  Thanks for this comment.  Weighted summation is a simple  yet effective way to  accomplish the adapter routing in LLMs. As shown in our appendix, the routing weights become shaper as the the network goes deeper.   We have also tried  two alternative schemes,  namely *mean routing* and *hard routing*, which achieves inferior performance than the weighted summation on ScienceQA.
>
> | ScienceQA          | Overall  Acc |
> | ------------------ | ------------ |
> | Mean routing       | 87.93        |
> | Hard routing       | 74.18        |
> | Weighted Summation | 89.41        |
>
> **Comment#4：**
>
> It seems the authors also introduce adapters to the image encoder. This should also be explained.
>
> **Response:**
>
>  Thanks for this insightful comment. This setting is also an important finding of our paper. To explain,  the adapters in image encoder can help to narrow the domain gap between the pre-trained and down-stream images.   For example, ScienceQA  includes a large number of synthetic images, which are  barely seen  by the pre-trained image encoder before, *i.e.,* CLIP-ViT.
>
>
>
> **Comment#5：**
>
> For the image input features, the authors use the `[CLS]` of every fourth layer of CLIP-ViT, are there any rationales behind this?
>
> **Response:**
>
>  Thanks for this careful review.   Using [CLS] features at different scales  can provide richer semantic information for multimodal tasks.  To explain, semantic information of visual features usually varies at different depths of the network. For example, shallow layers typically encode low-level semantic, such as texture and color. In this case, the use of visual features from different layers will facilitate the learning of attribute words.
>
>
>
> **Comment#6：**
>
> I'm confused about Sec. 3.2. It seems that Sec. 3.2 can be cohesively organized with that of MMT. Are there more considerations about this section? If not, the writing should be better organized.
>
> **Response:**
>
> Thanks for your suggestion. Sec. 3.2 mainly describes the model details of LaVIN, which can be combined together  with the descriptions of MMT. Following your suggestion, we will re-organize this section to improve the readability.
>
>
>
> **Comment#7：**
>
> - Some typos:
>   - Line 156, `m` and `n` are in reverse order;
>   - Line 170, the dimension definition is not correct.
>
> **Response:**
>
> Thanks for your careful review  and  we will revise these typos in our final version.

---

> > ### Comment · Reviewer_4c4u · 2023-08-22
> >
> > I have no further concerns and would like to keep my original `weak accept` score.

---

### Official Review · Reviewer_mD9H · 2023-07-04

**Soundness:** 3 good
**Presentation:** 3 good
**Contribution:** 2 fair
**Rating:** 4
**Confidence:** 5

**Summary:**

This paper presents a cost-efficient method to fine-tune LLMs thus enabling their multimodal reasoning capabilities. The main technical contribution includes using Mixture-of-Modality Adapation, which adopts lightweight adapters to bridge the gap between modality gaps. In the meanwhile, MMA also allows automatic routing such that the model can process both multimodal prompts and text-only prompts. Experiments show superior results on ScienceQA and training efficiency, in terms of both training time and number of trainable parameters.

**Strengths:**

- The proposed method may help significantly reduce the fine-tuning cost of LLMs with multimodal inputs.
- The proposed method is conceptually concise and generic. It could be potentially incorporated into different models/systems thus harvesting the development of both LLMs and vision encoders.
- Results on ScienceQA is promising, surpassing some strong competitors such as GPT-4.

**Weaknesses:**

- Evaluation is not sufficiently convincing. Only quantitative results on ScienceQA are presented. This leads to a narrowed view of the multimodal capabilities of the fine-tuned model. For example, does the model perform equally well on image captioning datasets such as NoCaps, COCO; how does the model perform on cross-modal retrieval / other VQA benchmarks? Missing these results makes it difficult to understand whether the proposed fine-tuning paradigm is actually bridging the modality gap as good as some previous works.
- In L50 and also Figure 1(b), authors claim "this paradigm often requires to update most parameters of LLM, limiting the efficiency of VL instruction tuning", "these fine-tune schemes will inevitably undermine the NLP capabilities of LLMs due to the drastic changes in their parameter spaces" and "existing multimodal LLM do not support text-only instructions, greatly hindering their applications". I wouldn't say these are correct. For example, BLIP-2/MiniGPT4 both keep LLMs frozen without updating their parameters, thus one can always prompt its LLMs with text-only prompt without degradation in language generation quality. I am therefore not fully convinced by the motivation of having automatic routing, instead of having just frozen LLMs speaking of to preserve the text generation capabilities.

**Questions:**

NA

**Limitations:**

Limitations are discussed.

---

> ### Author Rebuttal · Authors · 2023-08-09
>
> We sincerely appreciate your time and efforts spent in this paper.  Below, we response to your concerns point by point.
>
> **Comment#1：**
>
> Evaluation is not sufficiently convincing. Results on image captioning and VQA should be reported.
>
> **Response:**
>
> Thanks for this  constructive comment.  Following your suggestion, we conduct more experiments on other VL benchmarks, including your suggested COCO Captioning and VQA2.0, and  zero-shot multimodal evaluation and zero-shot question answering.    For a fair comparison, we follow the settings of previous works[B,18,37] on these benchmarks.
>
> | Image Captioning *Karpathy test* | Pre-training Data | Updated Params | BLEU@4 | Cider |
> | -------------------------------- | :---------------: | :------------: | :----: | :---: |
> | ClipCap [A]                      |         0         |       -        |  33.5  | 113.1 |
> | LLaMA-Adapter V2 [B]             |         0         |      14M       |  36.2  | 122.2 |
> | LaVIN (ours)                     |         0         |      5.4M      |  36.4  | 126.9 |
> | BLIP [16]                        |        14M        |      583M      |  40.4  | 136.7 |
> | BLIP-2 [15]                      |       129M        |      188M      |  43.7  | 145.3 |
> | LaVIN (ours)                     |       0.6M        |      5.4M      |  37.8  | 131.7 |
>
> | VQAv2 *val*          | Overall | Number | Yes/no | Other |
> | -------------------- | :-----: | :----: | :----: | :---: |
> | LLaMA-adapter V2 [B] |  67.22  | 49.34  | 84.83  | 56.59 |
> | LaVIN (ours)         |  68.74  | 51.12  | 87.67  | 59.01 |
>
> | Zero-shot ThruthfulQA [E] | Acc  |
> | ------------------------- | :--: |
> | LLama [37]                | 38.7 |
> | LLaVA [18]                | 16.4 |
> | LLaMA-Adapter V2 [B]      | 24.4 |
> | LaVIN (ours)              | 47.9 |
>
> | Zero-shot MME benchmark [F] | Cognition | Perception | Overall |
> | --------------------------- | :-------: | :--------: | :-----: |
> | BLIP2 [15]                  |  1293.84  |    290     | 1583.84 |
> | LLaMA-adapter v2 [B]        |  972.67   |   248.93   | 1221.6  |
> | MiniGPT-4 [48]              |  866.58   |   292.14   | 1158.72 |
> | LLaVA [18]                  |  502.82   |   214.64   | 717.46  |
> | LaVIN (ours)                |  963.61   |   249.64   | 1213.25 |
>
> From the above tables, we can observe some important findings.
>
> 1. As as an parameter-efficient tuning method, our LaVIN are consistently better than LLaMA-Adapter v2 under both supervised and zero-shot settings. For example, LaVIN outperforms LLaMA-adapter V2 by +1.52% on VQAv2 and +23.5% on zero-shot TruthfulQA, respectively.
> 2. Compared with BLIP and BLIP-2 pre-trained on large-scale VL data, our performance is still competitive, while the expenditure is much cheaper. **Notably, our tuning only takes 4 GPU hours on 8 A100s, while BLIP-2 requires more than 300 GPU hours on 16 A100s.**
> 3. Our  NLP ability also outperforms existing methods.   On TruthfulQA, we can see that  their zero-shot performance is  obviously  inferior to the original LLaMA.   In stark contrast,  LaVIN can further improve the performance by +8.3% than LLaMA through its mixture-of-modality adaptation.
>
> Overall, we believe that these results can further validate the effectiveness and generalization ability of LaVIN.  Following your suggestion, we will  supplement these results and discussions to our final version.
>
>
>
> **Comment#2：**
>
> The claiming about  "the NLP capabilities declines in fully tuned multimodal LLM'' are not correct, since BLIP-2/MiniGPT4 both keep LLMs frozen without degradation of NLP ability.  I am therefore not fully convinced by the motivation of automatic routing, instead of having just frozen LLMs.
>
> **Response:**
>
> Thanks for your careful review. We agree that MiniGPT-4 and BLIP-2 are inappropriate  examples  to support the argument of  "full VL  fine-tuning".    Nevertheless, their shortcomings lie in their expensive parameter and training costs, as discussed in Line 49-50 of our paper. For instance, BLIP-2 requires a deep neck branch to connect the vision and language models, which needs to be pre-trained on massive VL data for over 200 hours on 16 A100s.  In stark contrast, our LaVIN can only update 5.4M parameters and require only about 2 hours on 8 A100s.
>
> Aside from the inappropriate examples, our argument about full VL tuning still stands. Full VL tuning is still a popular paradigm to adapt LLMs to VL tasks [18,G,H,I,J], and it does greatly undermine the NLP capabilities of the used LLMs. For instance, after being fully tuned on VL data, the performance of LLaVA greatly declines on the zero-shot TruthfulQA, a NLP benchmark, especially compared with the default LLaMA.
>
> Meanwhile, the common adapter-based schemes are also sub-optimal. When using the adapter  VL data to handle examples of mixed modalities, its performance is still greatly affected on NLP evaluations, see the following table.  Besides, the performance of existing adapter-based methods like LLaMA-Adapter significantly lags behind that of fully tuned approach, *i.e.,* LLaVA, see our comparison on ScienceQA.
>
> Compared with the above paradigms, our MM-Adaption  provides an elegant  yet effective way to automatically handle the inputs of mixed modalities, *i.e.,* dynamic routing.  **Notably,  on zero-shot TruthfulQA, our NLP performance is even obviously better than the frozen LLaMA.**
>
>
>
>
> [A] ClipCap: CLIP Prefix for Image Captioning
>
> [B] LLaMA-Adapter V2: Parameter-Efficient Visual Instruction Model、
>
> [E] TruthfulQA: Measuring How Models Mimic Human Falsehoods
>
> [F] MME: A Comprehensive Evaluation Benchmark for Multimodal Large Language Models
>
> [G] Otter: A Multi-Modal Model with In-Context Instruction Tuning
>
> [H] Shikra: Unleashing Multimodal LLM’s Referential Dialogue Magic
>
> [I] LLaVAR: Enhanced Visual Instruction Tuning for Text-Rich Image Understanding
>
> [J] GPT4RoI: Instruction Tuning Large Language Model on Region-of-Interest

---

> > ### Comment · Reviewer_mD9H · 2023-08-17
> >
> > Thanks for the detailed response and additional results.
> >
> > The fact that LAVIN significantly underperforms BLIP-2 (and on some datasets even BLIP) diminishes its contributions, despite with fewer computation cost and data required. It may also be possible that these full fine-tuning models can quickly reach comparable performance to LAVIN with reduced data and compute needed.
> >
> > I'd better appreciate the parameter-efficient training methods if they manage to at least come close to full model tuning results. Otherwise I doubt their values in advancing the field. I decide to maintain the original recommendation as such.

---

> > > ### Author Response · Authors · 2023-08-17
> > >
> > > Thanks for your reply.   Actually,  under comparable settings,  LaVIN can already achieve comparable performance with the fully tuned model, i.e., LLaVA, on ScienceQA.   However, the performance of LaVIN still lags behind BLIP-2  due to the lack of large-scale pre-training, which is not the main focus of this paper.  We believe that large-scale pre-training  will greatly boost the performance of LaVIN.
> > >
> > > Besides, our contribution is actually orthogonal to BLIP-2.  Our mixture-of-modality adaptation can also apply to BLIP during its fine-tuning stage. As discussed in our rebuttal, frozen LLMs are still suboptimal for multimodal adaption.  With the help of LaVIN,  the NLP ability of BLIP-2 can be further improved.
> > >
> > > Based on these aspects,  we believe that LaVIN can still be an efficient yet effective multi-modal adaptation strategy for LLMs, which is indeed valuable to the community.

---

### Official Review · Reviewer_mWDr · 2023-07-06

**Soundness:** 4 excellent
**Presentation:** 4 excellent
**Contribution:** 3 good
**Rating:** 6
**Confidence:** 4

**Summary:**

> Update: I bumped up my rating to 6 after rebuttal

This paper proposes, LaVIN, an efficient and effective vision-language instruction tuning scheme to adapt LLMs. Specifically, the authors utilize parameter-efficient modules to adapt the LLaMA LM – they insert several adapters to the image encoder and mixture of modality adapters to the LM, the LM is expected to automatically select and route through the adapters of different modalities. LaVIN can be trained in an end-to-end fashion. Because only adapters are learned during training, LaVIN training is much more efficient than LLaVA that uses full tuning. Instruction tuning in LaVIN includes both text-image data and text-only data in a multi-task fashion (but with better separation due to adapters) to enable text-image instructions and text-only instruction at inference time. Strong results are achieved on ScienceQA.


**Strengths:**

1. The proposed method is simple and sound. By the use of a bunch of adapters, the model can naturally learn different-modality instructions in an end-to-end fashion.
2. LaVIN is much more efficient than LLaVA.
3. Empirical results on ScienceQA are strong.
4. The paper is well-written.


**Weaknesses:**

1. I think the proposed method resembles LLaMA-adapter a lot, maybe the authors should better note their difference early in the paper – currently LLaMA-adapter is not described until the experiment.
2. The experimental comparison with LLaMA-Adapter is not an apple-to-apple comparison because  LLaMA-Adapter and LaVIN use different instruction tuning datasets – the performance gap may simply originate from the instruction tuning datasets. A better baseline should be using LLaMA-Adapter on the same instruction tuning datasets.


**Questions:**

As mentioned in the weakness section, I think LLaMA-adapter should be described early in the paper and a fair comparison with it is needed.


**Limitations:**

The authors have discussed the limitations of the proposed method in Section 5.

---

> ### Author Rebuttal · Authors · 2023-08-09
>
> We highly appreciate your time and effort in reviewing this paper, and also thanks for  positive rating and beneficial feedback. Below, we response to your key concerns point by point.
>
> **Comment#1：**
>
> I think the proposed method resembles LLaMA-adapter a lot, maybe the authors should better note their difference early in the paper – currently LLaMA-adapter is not described until the experiment.
>
> **Response:**
>
> Thanks for your suggestion.   The main differences between our work and LLaMA-adapter lies in the following aspects:
>
> 1.   LLaMA-adapter is a static multi-modal approach, while our MM-Adapter can dynamically adjust its adaption across single-and multi-modal tasks.   This  dynamic routing design can not only help LLMs retain the NLP ability  but also facilitate the cross-modal instruction tuning, see the following table..
> 2.  In our mixture-of-modality adaptation (MMA), we  reveal the importance of end-to-end optimization for multi-modal LLMs,  *i.e.,* adding more adapters into visual backbone,  and also adopts the mixed modality training to improve the cross-modal training.   These innovative designs are also greatly different from LLaMA-adapter.
>
> Based on these differences, we think that our contribution is orthogonal to LLaMA-adapter.   Meanwhile, to our best knowledge, our MMA is currently the most efficient transfer learning scheme for LLaMA, which only takes 1.4 GPU hours to adapt to ScienceQA.
>
> Following your suggestion, we will highlight the above differences in our new version.
>
> | Zero-shot ThruthfulQA [E] | Acc  |
> | ------------------------- | :--: |
> | LLama [37]                | 38.7 |
> | LLaVA [18]                | 16.4 |
> | LLaMA-Adapter V2 [B]      | 24.4 |
> | LaVIN (ours)              | 47.9 |
>
> | Zero-shot MME benchmark [F] | Cognition | Perception | Overall |
> | --------------------------- | :-------: | :--------: | :-----: |
> | BLIP2 [15]                  |  1293.84  |    290     | 1583.84 |
> | LLaMA-adapter v2 [B]        |  972.67   |   248.93   | 1221.6  |
> | MiniGPT-4 [48]              |  866.58   |   292.14   | 1158.72 |
> | LLaVA [18]                  |  502.82   |   214.64   | 717.46  |
> | LaVIN (ours)                |  963.61   |   249.64   | 1213.25 |
>
> **Comment#2：**
>
> The experimental comparison with LLaMA-Adapter is not an apple-to-apple comparison because LLaMA-Adapter and LaVIN use different instruction tuning datasets – the performance gap may simply originate from the instruction tuning datasets. A better baseline should be using LLaMA-Adapter on the same instruction tuning datasets.
>
> **Response:**
>
> Thanks for this  suggestion.    In fact, the experimental comparison on ScienceQA and Multi-modal Chatbot are relatively fair  for LLaMA-Adapter and our LaVIN.   Particularly, both models are trained from scratch on ScienceQA and use the same text instruction data on Multi-modal Chatbot.
>
> To further address your concerns, we also supplement the comparisons on COCO Captioning and VQA2.0, of which experimental settings are the same for both methods. It can be seen that LaVIN is consistently better than LLaMA-Adapter in both performance and efficiency.
>
> | COCO Captioning      | Updated Params | BLEU@4 | Cider |
> | -------------------- | -------------- | ------ | ----- |
> | ClipCap [A]          | -              | 33.5   | 113.1 |
> | LLaMA-adapter V2 [B] | 14M            | 36.2   | 122.2 |
> | LaVIN (ours)         | 5.4M           | 36.4   | 126.9 |
>
> | VQAv2                | Overall | Number | Yes/no | Other |
> | -------------------- | ------- | ------ | ------ | ----- |
> | LLaMA-adapter V2 [B] | 67.22   | 49.34  | 84.83  | 56.59 |
> | LaVIN (ours)         | 68.74   | 51.12  | 87.67  | 59.01 |
>
> [A] ClipCap: CLIP Prefix for Image Captioning
>
> [B] LLaMA-Adapter V2: Parameter-Efficient Visual Instruction Model

---

> > ### Comment · Reviewer_mWDr · 2023-08-15
> >
> > Thanks for your detailed response! The added results on additional datasets are very helpful. Can you clarify one further question:
> > Regarding the backbone models of LLaMA-adapter and LaVIN, can you confirm that the image encoder and language decoder in LLaMA-adapter and LaVIN are exactly the same?

---

> > > ### Author Response · Authors · 2023-08-15
> > >
> > > Thanks a lot for your reply. Yes, LaVIN and LLaMA-V2 use the same image encoder, i.e., ViT-L/14 from CLIP. The text decoder is also kept the same in most experiments. i.e., LLaMA-7B, expect for zero-shot MME,  where LLaMA-13B is used in LaVIN.

---

> > > > ### Comment · Reviewer_mWDr · 2023-08-15
> > > >
> > > > How about LLaMA-Adapter V1 in the scienceQA results? Are the encoder/decoder also the same as LaVIN there?

---

> > > > > ### Author Response · Authors · 2023-08-15
> > > > >
> > > > > Yes, the image encoder and text decoder are exactly the same with LLaMA-Adapter on ScienceQA, i.e., ViT-L/14 from CLIP and LLaMA-7B.

---

> > > > > > ### Comment · Reviewer_mWDr · 2023-08-15
> > > > > >
> > > > > > Cool, I am happy to bump up my score to 6 then. Please add the additional experiments in the rebuttal to the next revision in the paper, they are very helpful.

---

> > > > > > > ### Author Response · Authors · 2023-08-15
> > > > > > >
> > > > > > > We will update these results in our final version.  Thanks again for your time.

---

### Official Review · Reviewer_q2tp · 2023-07-07

**Soundness:** 3 good
**Presentation:** 3 good
**Contribution:** 3 good
**Rating:** 7
**Confidence:** 3

**Summary:**

This paper proposes Mixture-of-Modality Adaptation (MMA), which adopts lightweight adapters to bridge the gap between LLMs and VL tasks. The adapter utilizes a router to automatic shift between single-modal and multi-modal instructions. When applying MMA to LLaMA and training on both single-modal and multi-modal data, their proposed approach achieves competitive performance with supervised methods on ScienceQA dataset. Besides, the training of MMA is efficient and cheap.

**Strengths:**

1. The mixture-of-modality adapter could dynamically adjust the adaptations for single-modal inputs and uni-modal inputs, which help preserve the NLP capability of LLMs.
2. Achieve competitive performance given small number of training parameters.
3. Ablation studies support the effectiveness of mixture-of-modality training and mixture-of-modality adaptation.

**Weaknesses:**

1. The adapter idea has been extensively explored in previous efficient VL training, and using adapter to efficiently bridge vision and LLM has been explored in LLaMA-Adapter.
2. Missing discussion/ablation of how the router in the MMA helps with the LLM learn visual information while preserving NLP capability.
3. This paper only evaluates on ScienceQA. Evaluation on other benchmarks like (COCO Caption) is needed for better comparison with LLaMA-Adapter.

**Questions:**

NA

---

> ### Author Rebuttal · Authors · 2023-08-09
>
> We sincerely appreciate your careful review and constructive suggestions for this paper. Below, we response to your key concerns point by point.
>
> **Comment#1：**
>
> The adapter idea has been extensively explored in previous efficient VL training, and using adapter to efficiently bridge vision and LLM has been explored in LLaMA-Adapter.
>
> **Response:**
>
> Thanks for this comment. Compared with existing adapters, the main difference of our mixture-of-modality adapter (MM-Adapter) lies in its dynamic routing property. Unlike previous works, MM-Adapter can automatically switch between single- and multi-modal tasks, which can not only retain the NLP ability of LLaMA but also facilitate the cross-modal instruction tuning , see table 2.
>
> In this paper, we also explore the mixture-of-modality training and reveal the importance of end-to-end optimization, *e.g.,* adding adapters for the visual backbone, which are also the key differences to LLaMA-Adapter.
>
> With these innovative designs, our LaVIN is obviously better than LLaMA-Adapter on various single- and multi-modal benchmarks， of which results are given in below.  Notably, to our best knowledge, our method is also the most efficient multimodal adaptation scheme for LLaMA, which only takes 1.4 hours to adapt to ScienceQA.
>
> | COCO Captioning      | Updated Params | BLEU@4 | Cider |
> | -------------------- | -------------- | ------ | ----- |
> | ClipCap [A]          | -              | 33.5   | 113.1 |
> | LLaMA-Adapter V2 [B] | 14M            | 36.2   | 122.2 |
> | LaVIN (ours)         | 5.4M           | 36.4   | 126.9 |
>
> | VQAv2 *val*          | Overall | Number | Yes/no | Other |
> | -------------------- | ------- | ------ | ------ | ----- |
> | LLaMA-adapter V2 [B] | 67.22   | 49.34  | 84.83  | 56.59 |
> | LaVIN (ours)         | 68.74   | 51.12  | 87.67  | 59.01 |
>
> | Zero-shot ThruthfulQA [E] | Acc  |
> | ------------------------- | ---- |
> | LLama [37]                | 38.7 |
> | LLaVA [18]                | 16.4 |
> | LLaMA-Adapter V2 [B]      | 24.4 |
> | LaVIN (ours)              | 47.9 |
>
> **Comment#2：**
>
> Missing discussion/ablation of how the router in the MMA helps with the LLM learn visual information while preserving NLP capability .
>
> **Response:**
>
> Thanks for this constructive comment.  As discussed above, the router in MMA can automatically choose the specific adapters for the single or multi-modal inputs, thus our MM-Adapters can learn visual information without contradicting to the default NLP capability of LLaMA.
>
> In MMA, we also adopt a mixture-of-modality training (MMT) regime to jointly train MM-Adapters on data of different modalities, which can further improve the NLP and VL capabilities of our LaVIN.  For instance, compared with  LLaMA-Adapter and LLaMA-Adapter v2, our LAVIN demonstrates consistently better performance on four benchmark datasets, including COCO captioning, VQAv2, ScienceQA and TruthfulQA.
>
> Following your suggestion, we will add the above discussions to our final submission.
>
>
>
> **Comment#3：**
>
> This paper only evaluates on ScienceQA. Evaluation on other benchmarks like (COCO Caption) is needed for better comparison with LLaMA-Adapter.
>
> **Response:**
>
> Thanks for this suggestion. The comparisons with LLaMA-adapter v2 on COCO captioning and VQA2.0 are given in the following tables.
> From these results, we can see that our LaVIN is consistently better than LLaMA-Adapter v2 in both performance and parameter efficiency.
>
> | COCO Captioning      | Updated Params | BLEU@4 | Cider |
> | -------------------- | :------------: | :----: | :---: |
> | ClipCap [A]          |       -        |  33.5  | 113.1 |
> | LLaMA-adapter V2 [B] |      14M       |  36.2  | 122.2 |
> | LaVIN (ours)         |      5.4M      |  36.4  | 126.9 |
>
> | VQAv2 *val*          | Overall | Number | Yes/no | Other |
> | -------------------- | ------- | ------ | ------ | ----- |
> | LLaMA-adapter V2 [B] | 67.22   | 49.34  | 84.83  | 56.59 |
> | LaVIN (ours)         | 68.74   | 51.12  | 87.67  | 59.01 |
>
> [A] ClipCap: CLIP Prefix for Image Captioning
>
> [B] LLaMA-Adapter V2: Parameter-Efficient Visual Instruction Model
>
> [E] TruthfulQA: Measuring How Models Mimic Human Falsehoods

---

### Official Review · Reviewer_3xnb · 2023-07-07

**Soundness:** 2 fair
**Presentation:** 3 good
**Contribution:** 3 good
**Rating:** 5
**Confidence:** 3

**Summary:**

The paper presents a novel method to do efficient vision language fine-tuning. Through a mixture of modality adaptation mechanism, the model can close the gap between different modalities. Additionally, the paper proposes a routing algorithm to switch between multiple
tasks. The training cost of the proposed system is low as the number of total trainable parameters is less than 4M. The model has been evaluated on public benchmark of ScienceQA.

**Strengths:**

- The proposed method is efficient. The total trainable parameters are less than 4M.
- The mixture-of-modality adaptation mechanism provides a way to adapt the LLM to vision modalities without expensive VL pretraining.
- The proposed method has been evaluated on public available benchmark, achieves comparable results, and for some of the evaluation metrics, it surpasses some existing models that have larger sizes

**Weaknesses:**

For weaknesses, please see my questions in below section.

**Questions:**

- The model is only tested on one dataset, which might not be convincing enough regarding the effectiveness of the model. Can the authors evaluate the model using more public datasets?
- The proposed model utilizes two adapters and one routing function to decide which adapter to use. The features generated after the routing function is affected by scale factor, will the performance be sensitive with the scale factor? Additionally, the routing function is
defined as a weighted summation, is there any other option that performs better?
- How to make it possible to extend the model to more modalities, e.g. video, audio signal.

---

> ### Author Rebuttal · Authors · 2023-08-09
>
> We highly appreciate your time and effort in reviewing this paper, and also thanks for your constructive comments on our work. Below, we response to your  concerns point by point.
>
> **Comment#1：**
>
> The model is only tested on one dataset, which ight not be convincing enough regarding the effectiveness of the model.
>
> **Response:**
>
> Thanks for your this comment.  Following your suggestion, we supplement a more comprehensive comparison of LaVIN  on  COCO Captioning [K],  VQA2.0 [L], TruthfulQA [E] and MME Benchmark [F]， and compare  LaVIN with three representative approaches, namely LLaMA-adapter V2 [B],  BLIP [16] and BLIP-2 [15].  The results are  reported in the following tables.
>
> | Image Captioning *Karpathy test* | Pre-training Data | Updated Params | BLEU@4 | Cider |
> | -------------------------------- | :---------------: | :------------: | :----: | :---: |
> | ClipCap [A]                      |         0         |       -        |  33.5  | 113.1 |
> | LLaMA-Adapter V2 [B]             |         0         |      14M       |  36.2  | 122.2 |
> | BLIP [16]                        |        14M        |      583M      |  40.4  | 136.7 |
> | BLIP-2 [15]                      |       129M        |      188M      |  43.7  | 145.3 |
> | LaVIN (ours)                     |         0         |      5.4M      |  36.4  | 126.9 |
> | LaVIN (ours)                     |       0.6M        |      5.4M      |  37.8  | 131.7 |
>
> | VQAv2 *val*          | Overall | Number | Yes/no | Other |
> | -------------------- | :-----: | :----: | :----: | :---: |
> | LLaMA-adapter V2 [B] |  67.22  | 49.34  | 84.83  | 56.59 |
> | LaVIN (ours)         |  68.74  | 51.12  | 87.67  | 59.01 |
>
> | Zero-shot ThruthfulQA [E] | Acc  |
> | ------------------------- | :--: |
> | LLama [37]                | 38.7 |
> | LLaVA [18]                | 16.4 |
> | LLaMA-Adapter V2 [B]      | 24.4 |
> | LaVIN (ours)              | 47.9 |
>
> | Zero-shot MME benchmark [F] | Cognition | Perception | Overall |
> | --------------------------- | :-------: | :--------: | :-----: |
> | BLIP2 [15]                  |  1293.84  |    290     | 1583.84 |
> | LLaMA-adapter v2 [B]        |  972.67   |   248.93   | 1221.6  |
> | MiniGPT-4 [48]              |  866.58   |   292.14   | 1158.72 |
> | LLaVA [18]                  |  502.82   |   214.64   | 717.46  |
> | LaVIN (ours)                |  963.61   |   249.64   | 1213.25 |
>
> From the above tables, we can observe some important findings.
>
> 1.  As as an parameter-efficient tuning method, our LaVIN are consistently better than LLaMA-Adapter v2 under both supervised and zero-shot settings. For example, LaVIN outperforms LLaMA-adapter V2 by +1.52% on VQAv2 and +23.5% on zero-shot TruthfulQA, respectively.
> 2.   Compared with BLIP and BLIP-2 pre-trained on large-scale VL data, our performance is still competitive, while the expenditure is much cheaper. For instance, with only 0.6M pre-training data and 5.4M updated parameters, LAVIN can achieve 131.7 CIDEr on COCO Captioning. **Notably, our tuning only takes 4 GPU hours on 8 A100s, while BLIP-2 requires more than 300 GPU hours on 16 A100s.**
> 3.   Our  NLP ability also outperforms existing methods. As discussed in the main paper, the  NLP capabilities  of most existing multimodal LLMs are  often undermined during VL instruction tuning.   On TruthfulQA, we can see that  their zero-shot performance is  obviously  inferior to the original LLaMA.   In stark contrast,  LaVIN can further improve the performance by +8.3% than LLaMA through its mixture-of-modality adaptation.
>
> Overall, we believe that these results can further validate the effectiveness and generalization ability of LaVIN.  Following your suggestion, we will  supplement these results and discussions to our final version.
>
>
>
> **Comment#2：**
>
> Will the performance be sensitive with the scale factor?
>
> **Response:**
>
> Yes, the selection of scale factor will  somewhat  affect the performance.  According to  LoRA [12],   the value of scale factor is positively related to the learning rate, but it does not need to be specifically tuned.
>
>
>
> **Comment#3：** Is there any other option for routing functions that performs better?
>
> **Response:**
>
>  Thanks for this insightful question.   In fact, we ever tried two alternative solutions for LaVIN, namely *mean-routing* and *hard-routing*.  In particular, mean-routing refers to the average output of the two adapters and is static for all examples. Compared with our MMA, hard-routing is also dynamic, but only selects one adapter at each inference step. This binary routing scheme is often more difficult to optimize [M].
>
> As shown in the following table, these  alternatives are obviously inferior to our MMA, *i.e.,* the weighed summation.
>
> | ScienceQA                 | Overall  Acc |
> | ------------------------- | :----------: |
> | Mean routing              |    87.93     |
> | Hard routing              |    74.18     |
> | Weighted Summation（ours) |    89.41     |
>
>
>
> **Comment#4：**
>
> How to extend the model to more modalities, e.g. video, audio signal ?
>
> **Response:**
>
> Yes, the extension of LaVIN can be summarized as the following steps:
>
> 1. Add  the backbones of the new modalities  for feature  embedding.
> 2. Mix the  training data of different modalities.
> 3. Add more routing paths in LLM  to adapt  different modalities.
> 4. Conduct multimodal training  via our mixture-of-modality training scheme.
>
> Based on these steps, we have successfully extended LaVIN to the video  data, which will be discussed  in our  next work.
>
>
>
> [A] ClipCap: CLIP Prefix for Image Captioning
>
> [B] LLaMA-Adapter V2: Parameter-Efficient Visual Instruction Model
>
> [E] TruthfulQA: Measuring How Models Mimic Human Falsehoods
>
> [F] MME: A Comprehensive Evaluation Benchmark for Multimodal Large Language Models
>
> [K] Microsoft COCO: Common Objects in Context
>
> [L] Making the V in VQA Matter: Elevating the Role of Image Understanding in Visual Question Answering
>
> [M] ProxylessNAS: Direct Neural Architecture Search on Target Task and Hardware

---

### Author Rebuttal · Authors · 2023-08-09

Dear Reviewers:



We thank all reviewers for their valuable and encouraging comments on the novelty and technical contributions of our paper, such as *"it surpasses some existing models that have larger size"*, *"achieve competitive performance given small number of training parameters "*, *"LaVIN is much more efficient than LLaVA"* and *" favorable results "*.   During the rebuttal phrase, our main responses include:

1.   Additional evaluations on new VL benchmarks, including COCO captioning, VQAv2, TruthfulQA and MME Benchmark, which  further validate the effectiveness and generalization  of  our method.

2. The extensive comparisons with LLaMA-Adpater on four additional benchmarks, which well confirm our superiority over this similar PETL methods.

3. The active discussion about the role of our cross-modality routing design, which is more convenient and effective than existing solutions towards handling the inputs of mixed modalities.

In addition, the key concerns of all reviewers are point-by-point addressed in each rebuttal.









Lastly, the codes of new experiments in our response are anonymously released at: https://anonymous.4open.science/r/LaVIN--1067, and the technical details can be found in the attachment.





Best,

The Authors

---

### Comment · Area_Chair_VDmx · 2023-08-18
**Respond to author rebuttals.**

Dear Reviewers 3xnb, q2tp, 4c4u,

Please respond to author rebuttals as soon as possible for timely communicating with authors until next Monday.

Thanks,
AC.

---

### Decision · Program_Chairs · 2023-09-21

**Decision:**

Accept (poster)

**Comment:**

This paper proposes a novel dynamic routing method with the mixture of adapters for efficient adaption of LLM for VL instruction model while maintaining the original NLP capability. Overall, there are two main concerns raised by reviewers: i) the lack of analysis on the proposed mixture of adapters and routing mechanism, which lead to efficient adaptation with such small number of parameters and even without additional VL training, ii) insufficient benchmark evaluation for VL and Language-only understanding tasks. During the rebuttal period, the authors have addressed these two concerns with more detailed explanation and additional experimental results, and 4 out of 5 reviewers have inclined to positive ratings. I agree with these reviewers that the concerns are somewhat resolved, and I recommend this paper to be accepted. However, the authors still need to address these two concerns more thoroughly, for example, more in-depth analysis (not just the final metric) on the benefits from the mixture and routing especially with the connection to the resulting efficiency, and more empirical validation on VL and Language-only understanding tasks. At this moment, it seems that this paper overstates the unnecessity of additional VL training by considering inferior results compared to BLIP-2.